# The Challenges of Tuberculosis Management beyond Professional Competence: Insights from Tuberculosis Outbreaks among Ethiopian Immigrants in Israel

**DOI:** 10.3390/tropicalmed9020029

**Published:** 2024-01-24

**Authors:** Hashem Bishara, Daniel Weiler-Ravell, Amer Saffouri, Manfred Green

**Affiliations:** 1Tuberculosis Clinic and Pulmonary Division, Galilee Medical Center, Nahariya, and Azrieli Faculty of Medicine, Bar-Ilan University, Safed 5290002, Israel; 2Pulmonary Division and Tuberculosis Clinic, Carmel Medical Center, Haifa 3498838, Israel; daniel@weiler.org.il; 3Tuberculosis Clinic and Internal Medicine, Nazareth Hospital, and Azrieli Faculty of Medicine, Bar-Ilan University, Safed 5290002, Israel; amer_saffouri@nazhosp.com; 4School of Public Health, Faculty of Social Welfare and Health Science, University of Haifa, Haifa 3498838, Israel; manfred.s.green@gmail.com

**Keywords:** tuberculosis outbreak, latent tuberculosis, preventive therapy, reception centers, Ethiopian immigrants, Israel, tuberculosis elimination

## Abstract

Controlling tuberculosis (TB) among immigrants from high-incidence countries presents a public health concern as well as a medical challenge. In this article, we investigate a TB outbreak in a community of people of Jewish descent who emigrated from Ethiopia to Israel (Israeli Ethiopians) that started in June 2022. The index case was a 20-year-old female who had recently immigrated to Israel with her family. Her pre-immigration tuberculin skin test was positive. After excluding active TB, treatment with daily isoniazid for latent TB (LTB) was started shortly after her arrival. A year later, she was diagnosed with smear-positive, culture-positive, pulmonary TB. Investigation of 83 contacts revealed five additional patients with active TB, and three of whom were members were of her household. In this article, we report the current TB outbreak, review previously published TB outbreaks involving Israeli Ethiopians, analyze the factors that triggered each of these outbreaks, and discuss the challenges that face the Israeli TB control program in an era of declining TB incidence and diminishing resources available for TB control.

## 1. Introduction

Tuberculosis (TB) was the leading cause of death globally from infectious diseases before the COVID-19 pandemic. According to the 2023 WHO report, TB notifications dropped during the first year of the COVID-19 pandemic but resurged with an estimated 7.5 million new cases of TB reported in 2022 compared with 5.8 million new cases in 2020 [1]. This drop in TB case notifications during the COVID-19 pandemic was related to disruptions in TB services. These included the reduced ability of patients to seek healthcare in the context of lockdowns and the reduced ability of the healthcare system’s to provide adequate TB diagnostic and treatment services [1,2].

Early diagnosis and efficient treatment of active and latent tuberculosis (LTB) cases are the mainstay of TB control and elimination [3,4,5,6]. To perform these tasks, TB control programs must rely on a committed professional healthcare staff and adequate resources and infrastructure to provide accessible and effective TB control services [3,4,5,6].

According to the Israeli Ministry of Health, there has been a 70% reduction in TB incidence in Israel during the last two decades from 6.9/100,000 in 2002 to 2.1/100,000 in 2022 (personal communication, Sevan Perl MD, TB department, Ministry of Health, Israel, 22 November 2023). TB incidence among native Israelis declined to 1/100,000. This low incidence of TB is an indicator of the effectiveness of the National Tuberculosis Program (INTBP) in Israel [7]. 

Despite this achievement, TB outbreaks still occur in Israel, involving a substantial number of active TB patients and their contacts [8]. These outbreaks impose a considerable burden on local TB control services, divert staff labor and resources, and hinder the efforts to eliminate TB. Therefore, investigating and preventing TB outbreaks are essential components of tuberculosis control programs, particularly in countries on the verge of TB elimination, such as Israel.

This paper aims to report an outbreak of TB among people of Jewish descent who immigrated from Ethiopia to Israel (Israeli Ethiopians), reviews the previously reported TB outbreaks involving this community in Israel, and discusses the factors that triggered these outbreaks and the challenges facing the national TB control program in Israel in an era of declining TB incidence and dwindling resources for TB control. 

## 2. Materials and Methods

### 2.1. Settings

The INTBP was launched in 1997 following a substantial increase in TB cases in Israel starting in the early 1990s that were caused by mass immigration from high-burden TB countries, mainly Ethiopia and the former USSR countries [9].

The program concentrated all TB services in nine specialist TB centers (TBCs), located according to regional needs, which provided all TB-related medical care in Israel [7]. This arrangement provided a critical mass of patients for each center, ensuring TBC staff would treat a sufficient number of TB patients to acquire and maintain the needed expertise and commitment to this task. The TBCs provided all means of diagnosis and treatment for active TB and latent TB. The INTBP relied on directly observed therapy (DOT) for all TB patients following WHO recommendations. TBCs were reimbursed according to a diagnosis-related fixed tariff for each individual treated. Ministry of Health officials regularly reviewed the TBCs’ performance and authorized TBC reimbursement.

All people of Jewish descent who emigrate from Ethiopia to Israel undergo pre-immigration screening for TB while still in Ethiopia [10]. The screening consists of a first-step tuberculin skin test (TST) and a chest X-ray. Those with symptoms that could be related to TB or an abnormal chest X-ray are examined clinically, and additional investigation is performed. Active TB cases are treated for at least two weeks before departure to Israel. Upon their arrival in Israel, Israeli Ethiopians are granted Israeli citizenship and housed in reception centers, and TB screening is continued with a second TST. For those with LTB, DOT with twice weekly isoniazid was provided for all Israeli Ethiopians housed in reception centers in the Zefat sub-district, and follow-up was provided by the regional TBC. However, because of logistic and financial issues, the DOT approach for LTB treatment ended in early 2016, thereafter LTB treatment has been provided in the form of daily self-administered isoniazid. Furthermore, because of the provincial nature of the Northern District of Israel and the considerable hardships faced by Israeli Ethiopians housed in reception centers, direct transportation to the TBC and back was provided.

### 2.2. Design

This was a retrospective study. We defined TB patients in this outbreak as definite TB cases if they were: Shared the same space or environment with an active infectious TB case in this outbreak (epidemiologically related);Had culture-positive TB with isolates matching the index case (IC) strain via genotyping.

Patients who were diagnosed by a TBC physician as TB patients but had negative TB cultures were considered non-definite TB cases.

Samples of sputum and pleural effusion fluids were stained by the Ziehl–Neelsen method. All samples were cultured for *Mycobacterium tuberculosis (M. tuberculosis)* at the National Mycobacterium Reference Laboratory using the BACTEC MGIT 960 system (Becton Dickinson Diagnostic Systems, Sparks, MD, USA). Species identification was performed using a commercial strip DNA probe test (Hain Lifescience GmbH, Nehren, Germany). Genotyping was performed using 43-spacer spoligotyping and a 24-locus mycobacterial interspersed repetitive-unit–variable-number of tandem-repeat (MIRU-VNTR) typing.

### 2.3. Contact Investigation

Contacts of the IC and contacts of all additional secondary TB patients diagnosed in this outbreak underwent the standard contact investigation guidelines [11,12]. 

Contact investigation was performed following a systematic inquiry to identify all contacts of infectious TB cases. These contacts were referred to the TBC to be evaluated for active and latent TB. The medical evaluation included a tuberculin skin test (TST) using the Mantoux method; those with a positive TST or a TST conversion (≥10 mm increase from the initial Mantoux result) were referred for a chest X-ray and further testing, if deemed necessary, to rule out active TB. After active TB was ruled out, these contacts were considered to have LTB infection. Daily isoniazid for 6–9 months was the standard first-line treatment offered to contacts with LTB, and monthly TBC physician follow-up appointments were scheduled.

### 2.4. TB Outbreaks Review

We searched the “PubMed” database for the terms “TB outbreak & Israel & Immigrant” and reviewed all reported TB outbreaks in which Israeli Ethiopians were involved. We defined a TB outbreak as whenever three or more epidemiologically related individuals were diagnosed and all had an identical *M. Tuberculosis* strain.

## 3. Results

### 3.1. The Current Outbreak

The IC was a 21-year-old female with a pre-immigration positive Mantoux test (Table 1). Following the exclusion of active TB, a six-month treatment regimen with daily self-administered isoniazid was started shortly after she arrived in Israel. However, a year later, she was hospitalized because of a persistent cough, weight loss, and low-grade fever. Her chest X-ray showed a pulmonary infiltrate with cavitation, and her sputum was smear-positive and culture-positive for *M. tuberculosis*. She was treated with a standard four-drug regimen for six months, leading to complete resolution of her symptoms and chest X-ray findings. At this point, it became clear that she had not taken the isoniazid treatment regularly and paused treatment if she felt fatigued, a symptom that she related to the isoniazid treatment.

The IC had 14 close contacts who underwent standard contact investigation as described previously. Three more TB cases were diagnosed through contact investigation among the IC’s household members. Two additional cases were diagnosed a few months later after these two patients actively sought medical attention because of TB-related symptoms.

Of the six TB cases diagnosed in this outbreak, five were culture-positive (definite cases), and one patient had culture-negative pleural TB (a non-definite case). The diagnosis in this non-definite case relied on clinical and laboratory findings, including a lymphocyte-predominant exudative pleural effusion with high levels of adenosine deaminase and a clinical response to treatment leading to full resolution of symptoms and findings. Overall, there were 89 contacts of the IC and the secondary TB cases, out of whom six contacts did not attend their scheduled appointment and were not examined at the TBC. 

### 3.2. Review of TB Outbreaks Involving Israeli Ethiopians

Our search of the PubMed database located 42 published papers matching our search; however, only 4 reports were relevant to our study [13,14,15,16]. Among these four publications, two papers were related to the same outbreak [13,14]. All four TB outbreaks (three historical and the current outbreak) were managed by the staff of the two TBCs located in northern Israel (Nahariya and Nazareth), where the authors (HB, AS) have been working and have been involved in the management of these outbreaks.

Table 2 summarizes the main characteristics of these four outbreaks and the factors that triggered them. Overall, there were 35 TB patients involved (range 6–13 TB patients/outbreak) and 673 contacts in these four outbreaks. 

The first TB outbreak among Israeli Ethiopians was reported in 2014 and involved six active cases [13]. The index case was a 21-year-old female diagnosed with smear-positive, culture-positive pulmonary TB while still in Ethiopia. She was treated for two weeks in Ethiopia before her departure to Israel, and her symptoms greatly improved. However, the diagnosis of active pulmonary TB was not reported promptly to the healthcare staff in Israel, and the patient did not inform the reception center staff in Israel of her medical condition. Therefore, upon her arrival in Israel, treatment for active TB was not resumed. By the time she was diagnosed with TB in Israel and the contact investigation was completed, five additional cases of active TB were diagnosed among her contacts. These five secondary TB cases were Israeli Ethiopians who underwent TB screening and had a normal chest X-rays while in Ethiopia shortly before they departed to Israel and, thus, TB had been excluded.

One such secondary case was the sister-in-law of the IC, a 28-year-old pregnant woman [14]. She had prolonged persistent respiratory symptoms without a fever. Her physician was reluctant to perform a chest X-ray because of her pregnancy and treated her with antibiotics, assuming she had an upper respiratory infection. Eventually, she was diagnosed with smear- and culture-positive pulmonary TB. Upon contact investigation, her two young daughters, aged 4 and 7 years, were also diagnosed with TB. Her husband was diagnosed with LTB infection. He was treated with isoniazid for five months, but treatment was discontinued because of isoniazid-induced hepatitis. The family moved shortly afterward to another district, and the husband’s LTB treatment was not resumed. Two years later, he was diagnosed with culture-positive peritoneal TB. Eventually, all four family members had active TB.

A second TB outbreak was also reported in 2014, involving 10 active TB patients and 174 contacts [15]. It originated from an incarcerated Israeli Ethiopian (source case, SC). The SC infection was transmitted to his cellmate, a 22-year-old Israeli male (IC), while both were imprisoned; however, neither was diagnosed by the prison healthcare services. Following the release of the IC, he transmitted the infection to his family and friends in a rural Arab-minority community. He moved into the basement of his parents’ house where his mother used to host a group of women from that neighborhood to have morning coffee together. This kind of “morning coffee gathering” is a common social activity among women in rural Arab communities. One of these women also brought her 2-year-old daughter with her. However, while the healthcare staff performed a classic systemic contact investigation, reaching out to all of the IC’s contacts, they were not aware of these women’s exposure during their morning gatherings. Thus, the woman and her 2-year-old daughter were not recognized as contacts and were not screened. Later on, the 2-year-old had a fever; she was hospitalized twice and discharged. Subsequently, she had convulsions and was hospitalized in a pediatric intensive care unit, but it took three days until she was diagnosed; sadly, she sustained severe neurologic damage.

The third outbreak was reported in 2020 and lasted nine years, involving 385 individuals, mainly second-generation Israeli Ethiopians [16]. This outbreak was exceptional in terms of its prolonged duration and low compliance with contact screening and LTB treatment. Out of 385 close contacts identified, 286 underwent contact investigation, 154 had a positive TST, 135 started LTB treatment, but only 26 completed treatment. 

## 4. Discussion

### 4.1. TB Outbreaks and Migration

This study examined TB outbreaks involving people of Jewish descent who emigrated from Ethiopia to Israel. 

TB outbreaks usually arise when the diagnosis of an infectious TB patient is delayed. Crowded living conditions of this community, in holding camps in Ethiopia before their departure and at the reception center following their arrival in Israel, enhance TB transmission.

Emigrants, in general, have an increased risk of developing TB due to prior infection as they mainly originate from high-burden countries. They also have barriers to accessing healthcare services, therefore have a greater risk their diagnoses being delayed, while they are living in crowded conditions that facilitate TB transmission [17,18,19]. 

However, with the implementation of pre-immigration TB screening and with full healthcare coverage in the host country, the prospects of people who emigrate from Ethiopia to Israel are much better than those of immigrants who do not enjoy such conditions [18,19,20,21]. 

TB outbreaks pose a substantial challenge to the healthcare system in terms of staff labor needed to cope with outbreaks, thus consuming much-needed human resources and a substantial cost to the healthcare system [22,23]. Thus, TB outbreaks are a setback for TB control efforts and may have significant health and financial repercussions for the individuals affected. Thus, it is relevant to investigate each TB outbreak and identify the factors that triggered it to prevent future outbreaks [24,25].

### 4.2. Insights from the Current TB Outbreak 

The current outbreak described in this report was the result of nonadherence to LTB treatment. Nonadherence to LTB treatment is a common reason for LTB treatment failure that increases the risk of TB disease [26,27]. However, healthcare staff are usually made aware of a patient’s unwillingness to comply with treatment once the patient does not attend their scheduled follow-up appointments. In such an event, the healthcare staff can try to persuade the patient to comply with treatment. However, this was not the case in this outbreak. The index patient arrived at her scheduled follow-up appointments and seemed to be compliant with treatment. However, in the aftermath, it became clear that she missed a few days of treatment now and then.

LTB treatment among Israeli Ethiopians in reception centers in the Zefat sub-district of northern Israel has been provided onsite by the nursing staff through the DOT approach for two decades [11]. Following the decline in TB patient numbers during the last few years and the subsequent shortage in TBC resources, the twice-weekly DOT approach for LTB treatment ended in 2016, and a daily self-administered LTB treatment approach was initiated instead. Therefore, in this case, it is likely that the termination of the DOT strategy for LTB treatment made it possible for the index case’s nonadherence to go unnoticed, triggering the TB outbreak.

Compliance with LTBI treatment is only one step in the cascade of care for TB and LTB patients [26,27]. This cascade starts with identifying those with a high risk of TB, screening those groups for TB and LTB, having them return for TST readings, and ensuring that those with a positive TST or interferon-gamma release assay (IGRA) receive further medical evaluation (physical examination and chest X-ray), and making the appropriate treatment decision. If LTB treatment is indicated, treatment is started and followed up, and patient compliance and treatment completion are ensured. Any breach in the above-mentioned stages of the cascade of care may bring about a new TB case and, in a congregated setting, a TB outbreak. 

### 4.3. Insights from the TB Outbreaks Review

The three previous TB outbreaks reviewed in this paper were all triggered by breaches in the TB cascade of care, mainly failures at the system level, which ultimately triggered a TB outbreak. 

The first 2014 outbreak was triggered by miscommunication between the medical staff involved in Israeli Ethiopian care, those in Ethiopia and those in Israel. The failure to promptly transfer relevant information regarding the recent diagnosis of active TB in the index case to healthcare personnel in Israel led to a pause in TB treatment and eventually, to the TB outbreak, which was further fueled by the crowded conditions in reception centers. The reluctance of the physicians to X-ray a high-risk pregnant woman with respiratory symptoms caused a delay in diagnosis facilitating transmission of TB disease in her community.

The second outbreak in 2014 illustrates how disregarding TB control measures in the prison healthcare system can facilitate TB transmission and triggere a TB outbreak in the community [28,29,30]. It also demonstrates the importance of maintaining functional TB control services that can cope with TB outbreaks. The TBC staff was downsized to half of its original workforce a year earlier and, therefore, could not cope promptly with the substantial number of cases and contacts. Downsizing the TBC staff in this case caused the contact investigation to progress at a slow pace, allowing the cascade of infection and disease to continue unimpeded by TB control intervention. One such contact who, at initial contact evaluation, had a normal chest X-ray, did not return to have his TST reaction measured, and six months later he was diagnosed with smear- and culture-positive pulmonary TB. Furthermore, the tragic case of TB meningitis in the 2-year-old in this outbreak could have been prevented had the healthcare staff been minded to the cultural practices in this rural community.

This outbreak illustrates the importance of applying TB control and prevention measures in correctional facility settings. It also illustrates how cutting down TBC staff and lack of familiarity with a rural community’s cultural practices and habits contributed to the outbreak’s extent and morbidity. 

The third (2020) outbreak was sustained by the low rate of compliance with TB contact screening and LTB treatment which eventually perpetuated the cycle of infection and disease in the affected community. Difficulties in obtaining transportation to the TBC had a deterrent effect, causing many contacts to drop out of follow-up. For some contacts, the arrival at the TBC was an onerous task because, without possessing a private car, there was no convenient way to reach it via public transportation. They had to take three buses just to get to the TB clinic and had to miss a day of work. Many contacts who arrived at the first appointment and had a TST performed did not return to have their TST reaction measured 48–72 hours later. Another factor that fueled this outbreak was the unwillingness of some TB patients to provide the identification details of their contacts, thus leaving many contacts unidentified and not included in the contact investigation. This behavior may have stemmed out of fear of being stigmatized. Eventually, incomplete contact investigation left a pool of unidentified contacts with latent TB infection untreated, some of whom went on to have active disease, thus sustaining the outbreak. Two of the cases in this outbreak were only retroactively related to it after genotyping results became available and the epidemiological relationship became known. Moreover, the possibility of a false-positive TST reaction among contacts who were BCG-vaccinated caused many contacts to doubt the validity of the positive TST results. Applying the interferon-gamma release assay (IGRA) test to investigate TB infection among active TB case contacts could filter out many otherwise false “positive” TST contacts. It could also spare the patients a second visit required with the TST, and may enhance compliance among those with a positive IGRA test [31,32]. This option is well justified in the case of Ethiopian Israelis who have a 40% prevalence of positive TST and whose LTB treatment requires them to endure an exhausting journey to the TBC and back.

This outbreak illustrates the importance of providing easy access to TB clinics in provincial areas. This could be achieved either through staff outreach or by providing convenient free-of-charge transportation to TB clinics. This outbreak also illustrates the need to involve social workers to help overcome TB-related stigmas and financial barriers in this community.

### 4.4. The Challenges of Tuberculosis Management in the Era of Diminishing TB Incidence

With the consistent decline in TB incidence and the number of TB patients in Israel, new challenges to TB control efforts are emerging, including financing TB control services and preserving adequate TB infrastructure and professional staff. 

#### 4.4.1. Financing TB Control Facilities

In Israel, TBCs operate as a fee-for-service organization. Therefore, the decline in TB patient numbers has led to a significant reduction in TBCs revenues. Meanwhile, the expenditures of these clinics have remained almost unchanged. This has made TBCs non-profitable and sometimes a financial burden. The TB clinics located in peripheral regions of Israel are the most affected. Subsequently this has led to the downsizing of the TBC staff, which proved to have a devastating impact on their capability to cope with large-scale TB outbreaks [15].

The fee-for-service model for TBC was adequate when there were substantial numbers of TB patients. The declining patient numbers and, consequently, TBCs revenues demonstrate a need to revise INTBP financial arrangements. 

The decline in TB patient numbers has also led to a significant reduction in the number of TB patients who need to be hospitalized. Subsequently, the two TB-dedicated hospital wards in Israel have been closed, leaving the most challenging TB patients without the option of being hospitalized in a TB-dedicated medical department. Treating drug-resistant TB patients, the homeless, drug addicts, and the mentally ill have become particularly challenging. These are the most vulnerable, underprivileged, hard-to-reach, and hard-to-treat TB patients. While such patients were previously treated in a TB-dedicated hospital setting, at least during the intensive phase of treatment, there are currently no easy solutions for such complicated TB patients. We believe that a dedicated TB ward should be kept operational, even if such a facility proves to be non-profitable. The merits of having a TB ward to accommodate the most complicated TB cases outweigh the cost considerations.

#### 4.4.2. Accessibility of TB Services

Concentrating TB care in nine dedicated TB centers in Israel allowed TBCs staff to gain expertise and professional competence in all aspects of TB care. However, it made access to TB care more difficult, particularly in the peripheral regions of Israel. We overcame this obstacle for Israeli Ethiopians housed in reception centers by reaching out and granting free direct transportation to the TBC. However, these measures were only granted to active TB patients as well as Israeli Ethiopians in reception centers. Other than these two groups, LTB-infected individuals have to make the journey to TBCs on their own. Transportation difficulties have proven to be a major barrier causing noncompliance and treatment dropouts, particularly among second-generation Israeli Ethiopians. 

#### 4.4.3. Updating TB and LTB Treatment Protocols

The “National TB Advisory Committee” (NTBAC) was instituted in early 1997. This forum conceptualized the INTBP and authored its medical and administrative sections. However, there have been some significant changes in the TB landscape since the inauguration of the INTBP. Therefore, it is time to revise the INTBP and adapt it to evolution in TB management. The advent of extensive drug-resistant TB (MDR-XDR) and the approval of new short-course treatment regimens for XDR-TB, and for LTB such as the three-month, once-weekly rifapentine-plus-isoniazid regimen. This once weekly short course treatment is an effective alternative to six months of isoniazid treatment option that may be better suited for Israeli Ethiopians in reception centers [33]. However, rifapentine is not yet available in Israel, as in other European countries [34].

## 5. Study Limitations and Strengths

Our study has some limitations: The small number of TB outbreaks in this review, all reported and managed by two provincial TBCs in northern Israel, limits the validity of our conclusions. Furthermore, the review focused on TB outbreaks involving Israeli Ethiopians; therefore, our conclusions are mainly limited to similar constellations and milieus and might not apply to TB outbreaks in other settings. However, we believe that there is much in common between TB outbreaks among Israeli Ethiopians and other immigrant populations in Western countries, and therefore, our insights may be relevant to these settings as well.

This paper also has some strengths. Two of the authors (HB, AS) were involved in managing all four outbreaks reviewed in this article and therefore, had first-hand knowledge of the details of each outbreak.

## 6. Conclusions

This paper highlights the paradoxical situation where success in dealing with an influx of active and latent TB subjects in a community led to deterioration of TB control and local resurgences of the disease. 

In this article, we discussed a recent TB outbreak among Israeli Ethiopians and reviewed previously reported TB outbreaks involving Israeli Ethiopians. We discussed the factors that contributed to these outbreaks and presented our insights as to the challenges facing the INTBP and the adaptations needed to be considered in TB control programs in an era of declining TB incidence.

With other Western countries on the verge of TB elimination, while facing the challenges of controlling TB among people who emigrate from high-burden TB countries, our insights may be of some relevance for such countries as well.

## Figures and Tables

**Table 1 tropicalmed-09-00029-t001:** Epidemiological and clinical characteristics of TB patients.

Pt N	Gen	Relation	Birth Date	Organ	Sm/Cult	Diagnosis M/Y	TST	Contacts Ident/Test
IC	F	Index case	04/2001	Lungs	+/+	02/2022	15/	14/14
1	M	IC’s brother	04/2005	Lungs	+/+	03/2022	0/18	7/7
2	F	IC’s Sister	04/2007	Lungs	−/+	03/2022	0/20	24/22
3	M	IC’s brother	04/1990	Lungs	−/+	03/2022	0/22	29/25
4	F	Pt1’s classmate	04/2005	Pleural	−/−	06/2022	0/16	None
5	F	IC’s workmate	04/2001	Lungs	+/+	10/2022	0/	15/15

Pt N, patient number; Gen, gender; Sm/Cult, smear/culture; M/Y, month/year; TST, tuberculin skin test pre/post-exposure; Contacts Ident/test, number of contacts identified/tested; IC, index case; F, female; M, male; Pt1, patient number 1.

**Table 2 tropicalmed-09-00029-t002:** TB outbreaks involving Israeli Ethiopians: characteristics and contributing factors.

Year	TB Cases	Cont. N.	Outbreak Contributing Factors	
			Patient Level	System Level
2022 *	6	89	Nonadherence to LTB treatment	Termination of DOT for LTB
2014 [12]	6	25	Patient miscommunication, crowded living conditions	Staff miscommunication
2014 [15]	10	174	Rural community cultural practices, noncompliance with contact screening, smoking	Prison TB control measures ignored, downsizing of TB clinic staff, staff unfamiliarity with rural community cultural practices
2020 [16]	13	385	Noncompliance with contact screening and LTB treatment, poverty, substance abuse, stigma	Challenging transportation route to the TB clinic, partial identification of TB contacts, unavailability of IGRA tests

TB, tuberculosis; Cont. N, contacts number; *, the current outbreak; LTB, latent tuberculosis; DOT, directly observed treatment; IGRA, interferon-gamma release assay.

## Data Availability

The data that support the findings of this study are available from the corresponding author (hashenb@gmc.gov.il) with the permission of the Nazareth Hospital.

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
