# Peer review of "The Challenges of Tuberculosis Management beyond Professional Competence: Insights from Tuberculosis Outbreaks among Ethiopian Immigrants in Israel"

_tropicalmed, 2024, doi:10.3390/tropicalmed9020029_

Round 1

Reviewer 1 Report

Comments and Suggestions for Authors

Major revisions:

1. Line 21 - the use of the term 'misleading' is pejorative. It blames the patient, who is by definition vulnerable, being a migrant in a period of crisis. This is compounded by the authors then say that "We believe that cessation of our standard policy of providing directly-observed Isoniazid treatment ... have contributed to this outbreak". I.e., that the challenges she experienced around adherence could / should actually be attributed to health system failures. Strongly suggest that the sentence starting at the end of line 20 and ending on line 22 ("However ...") is unnecessary and should be excluded. Similarly, the tone of lines 125-127 and line 131.

2. The acronym "EI" is unnecessary and serves to distance the authors and readers from the humanity of the people who bore the brunt of this outbreak. In many parts of the world, including in Israel, immigration is highly contentious and emotionally evocative. Migrants affected by TB often experience inferior health service access, stigma, and poorer outcomes. With that context, we should do all we can to keep our readers aware of the humanity and people who are affected. I strongly suggest removing the "EI" acronym and revising this language throughout. For example, line 58 could read: "This paper aims to report an outbreak of TB among people emigrating from Ethiopia in reception center in Israel", or lines 89-90 could read: "... was provide in reception centers for all people immigrating from Ethiopia".

3. Lines 224-226 - Why is transmission between two people who shared a confined space remarkable? And both in prison, so by definition vulnerable. Why would ethnicity be relevant here. Yes, there are some genetic interactions between host and TB that increase the likelihood of infection and progress to disease, but these are far outweighed by exposure and socio-economic vulnerability. The emphasis on transmission between ethnic groups reifies differences between people and obfuscates the underlying drivers of transmission / disease; i.e., poverty, vulnerability, inadequate health service access.

4. Overall structure of the manuscript, especially the findings section - at current, the manuscript first presents a 'new' outbreak and then a review of three preceding outbreaks. This is done as separate tasks / components of the manuscript. Further, the 'lessons learnt' from each outbreak is limited to that outbreak, rather than cross-cutting lessons from all of the outbreaks considered together. Firstly, it is challenging for the reader to consider two designs in the same manuscript. I suggest that instead the authors could present this as a review of four outbreaks, three of which have only historical data, and one which they have current / contemporary data. Then, the design becomes singular and more coherent, a review of outbreaks over time to identify patterns. Secondly, the authors can then harmonise their interpretation of common and exceptional factors influencing outbreak progression across the four outbreaks. This would be similar to a thematic analysis. Three key ideas appear to be (1) lack of coordination between the country of origin and host country health services, (2) social vulnerability of the migrants impeding their ability to complete care, and (3) health service delivery challenges, especially challenges with contact tracing. Thirdly and finally, these themes / lessons could then be more easily compared with what is already known about challenges around immigration / cross-border TB control (e.g., see recent literature on TB in Ukranian migrants, or long-established patterns around TB and migration in Africa) to show what is novel about the findings.

5. The current content of the discussion is not appropriate. It is a 'discussion' of the authors' views on why the current TB outbreak occurred. Instead, it should be a discussion of how the findings presented in this manuscript compare with what is already known in the literature about TB and migration. I strongly suggest that much of the current content be shifted to the findings / results section, with a stronger academic discussion substituted as the "Discussion" section. 

Minor comments:

6. Line 94 - what does "epidemiologically related" mean? Suggest instead that the content of lines 97-98 is the information needed here: "people sharing the same space or environment with an active infectious TB case".

7. Line 113 - how successful was the contact investigation in this population? On the one hand, they are dependent on the new host state (Israel) and may be living in a contained geographic space, facilitating contact investigation. On the other hand, they may be reluctant to cooperate with the contact investigations if they fear (even in error) that being diagnosed with TB may jeopardize their immigration status. Some context would be useful for the reader.

8. The conclusions should be more specific to making recommendations for policy and practice in Israel. The findings are available to do so. What should be done to better serve people migrating from Ethiopia to Israel and to prevent future outbreaks?

Author Response

We wish to thank the reviewer for the valuable comments and suggestions that helped us to improve the manuscript. We appreciate the time and effort that the reviewers have dedicated to providing their insightful comments regarding this manuscript. We have been able to incorporate changes to reflect most of the suggestions provided by the reviewers.

Reviewer 2 Report

Comments and Suggestions for Authors

Interesting & relevant paper describing TB control in the setting of insufficient resources and importance of both good history-taking and understanding vulnerable communities.

The authors should comment on Rifapentine & INH for 12 weekly doses for TB prevention as could be more cost effective (Sterling NEJM 2011; 365:1607). Urine could be screened for INH metabolites

Comments on the Quality of English Language

Quality ok for colleagues whose 1st language is not English (see suggested edits)

Author Response

(The authors gave the same response as above.)

Reviewer 3 Report

Comments and Suggestions for Authors

1. If the objective of the work is to present a narrative review of the cases of TB in Israel and then the presentation of the case in question, I suggest a rewrite in the order of presentation. 

The chapter "results" should be divided into 2 parts: part 1 "Narrative review - results" and part 2 "Case report" or "Outbreak study".

While the description of the methodology of the review narrative should be in the materials and methods, suggesting that in the M&M there are also 2 parts - those relating to the narrative review and the others to the case report.

2. In the abstract and in the objectives of the work should be stated the fact that this article has 2 phases: 1. narrative review and 2, case report.

3. line 49 "(personal communication, Sevan Perl MD, TB department, Ministry of 49 Health, Israel)" should be a normal reference, in this case, the number 7. And in the bibliografic list must appear the date too.

4. A discussion has a lack of references to harmonize or text. I suggest increasing the number of references in the discussion.

5. Besides this, I would only consider small revisions of English.

Comments on the Quality of English Language

small revisions of English needed.

Author Response

(The authors gave the same response as above.)

Reviewer 4 Report

Comments and Suggestions for Authors

Minor Comments for the authors

Point 1: What cultural and social factors contribute to noncompliance with contact investigation and treatment among Ethiopian immigrants?

Point 2:Why is directly observed treatment important for preventing future outbreaks among Ethiopian immigrants?

Point 3: What are the challenges in managing TB outbreaks among Ethiopian immigrants in Israel?

Point 4: It Would be better to draw one extra figure of a detailed workflow of the study. Otherwise, it would be difficult for the reader to capture the overall picture of the study.

Point 5: Overall I could not fault the experiments or interpretationFuture studies should investigate the impact of diminishing resources allocated to TB control programs in Israel on the management and prevention of TB outbreaks among Ethiopian immigrants. Further research is needed to explore the specific cultural and social factors that contribute to noncompliance with contact investigation, unwillingness to provide contact details, difficulties with public transportation to TB clinics, poverty, and substance abuse among Ethiopian immigrants in Israel.

Good Luck

Comments on the Quality of English Language

Minor editing of English language required

Author Response

We wish to thank the reviewer for the valuable comments and suggestions that helped us to improve the manuscript. We appreciate the time and effort that the reviewers have dedicated to providing their insightful comments regarding this manuscript. We have been able to incorporate changes to reflect most of the suggestions provided by the reviewer.

Reviewer 5 Report

Comments and Suggestions for Authors

The reviewed article addresses the extremely topical issue of infectious disease control in the context of migration. Migration is becoming a challenge for an increasing number of countries and regions, and the shortcomings described in the paper regarding the control of infectious diseases that can accompany migrants are universal and do not apply only to the cited situation of Ethiopian migrants going to Israel. Three basic tasks to improve the prevention of the spread of infectious outbreaks were highlighted: prompt and proper communication between the various medical units associated with handling migrants, discernment of the traditions and habits of the various ethnic groups, and accessibility to medical facilities providing assistance to migrants. In the reviewer's opinion, the paper can be published in its current form. 

Author Response

We wish to thank the reviewer for the kind comments

Round 2

Reviewer 1 Report

Comments and Suggestions for Authors

Congratulations to the co-authors on this important work. You have addressed my previous comments satisfactorily. I wish you well in the continued work on this topic.